

# Development of simple sequence repeat markers and genetic diversity of *Sipunculus nudus* in the Beibu Gulf of China

Chunli Han[1,*], Yuzhu Ni[1,*], Guohao Yang[1], Jialin Yang[1], Jie Zou[2], Huijing Peng[2] and Pengliang Wang[1]

[1] Guangxi Key Laboratory of Marine Environment Change and Disaster in Beibu Gulf, Pinglu Canal and Beibu Gulf Coastal Ecosystem Observation and Research Station of Guangxi, College of Marine Sciences, Beibu Gulf University, Qinzhou, Guangxi, China

[2] Guangxi Institute of Oceanology, Beihai, Guangxi, China

[*] These authors contributed equally to this work.

Corresponding authors
Huijing Peng, 165080921@qq.com
Pengliang Wang,
pengliang_wang@163.com

## ABSTRACT

**Background**. The marine species *Sipunculus nudus*, valued for its medicinal and commercial significance, faces critical research gaps due to a lack of molecular markers (notably simple sequence repeats (SSRs)) and insufficient genetic diversity data, hindering genetic studies and evidence-based breeding initiatives.

**Methods**. The software of Misa and Primer3 were adopted to detect SSRs and develop primer pairs, and then some primers of highly polymorphic loci in the genome were used to reveal the genetic diversity of *S. nudus* along the Beibu Gulf in China.

**Results**. From the genome, 277,264 SSRs were detected an d 198,902 primers were designed. Ninety-five of the synthesized 100 primers could amplify across samples, with 93 showing polymorphism. Thirty highly polymorphic primers were chosen to disclose the genetic diversity of 219 materials from Beibu Gulf, China. We detected 370 alleles with high genetic diversity: observed heterozygosity ($Ho$) = 0.905, expected heterozygosity ($He$) = 0.833, Shannon's index ($I$) = 2.042, and polymorphism information content ($PIC$) = 0.811. The study also revealed varying levels of genetic differentiation and gene flow among different provenances. Structure analysis partitioned all samples into seven distinct genetic clusters.

**Conclusions**. We identified 277,264 SSRs in *S. nudus* and developed primers for 198,902 SSRs. Subsequently, 100 primers were validated for assessing genetic diversity and population differentiation. These will establish a critical foundation for advancing germplasm conservation and targeted breeding strategies in this species.

## INTRODUCTION

*Sipunculus nudus* is an economically important invertebrate species belonging to the family *Sipunculidae*. It is distributed along all coastal areas of China, and especially abundant in the coastal waters of Beibu Gulf in China (*Chen et al., 2008*; *Li et al., 2017*).

Modern medical researches have reported that *S. nudus* demonstrated multiple therapeutic effects including anti-fatigue (*Liu et al., 2012*), promoting wound healing (*Zhang, Dai & Cai, 2011*), enhancing cellular immunity (*Cao, 2023*), delaying aging (*Shen et al., 2004*), lowering blood pressure (*Cai et al., 2023*) and antioxidant activity (*Zhang & Dai, 2011*; *Li et al., 2016*). *S. nudus* was also known as "sea medicine" in ancient times. Additionally, *S. nudus* was also valued as a delicacy for its exquisite flavor. Rising living standards and health awareness boosted demand for *S. nudus*, causing overfishing that rapidly depleted populations and reduced genetic diversity (*Chen, Dong & Xu, 2020*). Hence, the monitoring of the genetic diversity of the natural resources of *S. nudus* are essential and urgent.

Genetic diversity serves as the foundation for adaption to the environment and genetic improvement of important traits (*Cui et al., 2012*). To date, current investigations into genetic diversity of *S. nudus* presented divergent findings across two research dimensions. Comparative analyses demonstrated significantly higher genetic variations in wild populations relative to their cultured counterparts (*Zhou et al., 2017*). Meanwhile, regional assessments showed elevated diversity levels in southern Chinese coastal populations (Fujian, Guangdong, Guangxi) through several molecular markers including mitochondrial fragments and RAPDs (*Du et al., 2009*; *Song et al., 2011*; *Ning et al., 2012*; *Peng et al., 2017*). However, contrasting reports indicated reduced genetic variation along certain Chinese coastal transects (*Song et al., 2017*). These discrepancies likely stemmed from two critical aspects: sampling strategy and selection of molecular markers.

Molecular markers are the useful tools to reveal genetic diversity. However, the molecular markers available in *S. nudus* included RAPDs (*Song et al., 2011*), few simple sequence repeats (SSRs) (*Guo et al., 2012*; *Wang et al., 2012*) and fragments of mitochondrial genome such as *COI* (the mitochondrial cytochrome coxidase subunit I) (*Ning et al., 2012*; *Hsu et al., 2013*), D-loop (*Zhou et al., 2017*; *Peng et al., 2017*), *cytb* (*Song et al., 2017*) and *16S* (*Qinghen et al., 2008*). The number of molecular markers in *S. nudus* were too limited to meet the demand of marker-aided breeding, although a genome sequence was documented (*Zheng et al., 2023*). Therefore, the genetic diversity of *S. nudus* were seriously hindered.

To date, SSR markers were considered to be the best useful in the molecular markers of the second generation. SSR markers possess the advantages including as co-dominance, multiple alleles, genome-wide distribution and high polymorphism (*Bagshaw, 2017*), contributing to better revealing genetic diversity and marker-aided breeding. Hence, large scale development of genome-wide SSR markers is essential, which will better reveal the genetic diversity and dramatically boost the marker-aided breeding of *S. nudus*.

In this study, our aims are (1) to develop a large number of genome-wide SSR markers and (2) systemically study the genetic diversity of *S. nudus* in the Beibu Gulf region in China.

## MATERIALS & METHODS

### Materials

A total of 219 samples of *S. nudus* were collected from eight geographic locations, namely Tieshangang District, Beihai City (TSG, 27 samples); Yingpan Town, Beihai City (YP, 28

**Table 1  The detailed information of samples in this study.**

| Number | Populations | Location | Latitude and longitude | Number of samples |
|---|---|---|---|---|
| 1 | TSG | Tieshan Port Area, Beihai City | 21.45N,109.46E | 27 |
| 2 | FCG | Guangpo Town, Fangchenggang City | 21.63N,108.54E | 27 |
| 3 | QZ | Rhinoceros Horn Town, Qinzhou City | 21.64N,108.73E | 29 |
| 4 | YP | Yingpan Town, Beihai City | 21.45N,109.41E | 28 |
| 5 | BH | Yinhai District, Beihai City | 21.25N,109.4E | 27 |
| 6 | HP | Shatian Town, Hepu County | 21.52N,109.64E | 27 |
| 7 | ZJ | Caotang Town, Zhanjiang City | 21.34N,109.75E | 27 |
| 8 | HN | Danzhou City, Hainan Province | 19.84N, 109.48E | 27 |

samples); Yinhai District, Beihai City (BH, 27 samples); Hepu County, Beihai City (HP, 27 samples); Rhinoceros Horn Town, Qinzhou City (QZ, 29 samples); Guangpo Town, Fangchenggang City (FCG, 27 samples); Caotan Town, Zhanjiang City (ZJ, 27 samples) and Guangcun Town, Danzhou City, Hainan Province (HN, 27 samples) (Table 1; Fig. 1). All samples were transported back to the laboratory. The body wall of each sample was washed with PBS and used to extract genomic DNA *via* TIANamp Marine Animals DNA Kit (DP324, Tiangen Biotech; Beijing; China).

## Development of genome-wide SSRs

The MIcroSAtellite (MISA) tool was utilized to identify SSR loci in the genome deposited in the Genome Warehouse in National Genomics Data Center (BioSample: SAMC4151241). The criteria for SSRs in this study were as follows: a minimum of 20, 12, six, five, four and three repeat units for one to six nucleotide motifs, respectively (*Han et al., 2024*). The primer3 was employed to design the corresponding primer pairs by default parameters.

A hundred primer pairs (15–18 primer pairs were selected for each motif type, with five to seven pairs evenly distributed on each chromosome) were selected and synthesized by Sangon Biotech (Shanghai) Co., Ltd.

## PCR amplification and polymorphic SSRs screening

The synthesized SSR primer pairs were screened using six random DNA samples as templates. PCR amplification was performed in a volume of 10 µL, which contained 2× SanTaq PCR Mix (B532061; Sangon Biotech (Shanghai) Co., Ltd.) five µL, 0.2 µL each of forward and reverse SSR primers (10 µmol L$^{-1}$), one µL of DNA template and 3.6 µL of ddH$_2$O (*Han et al., 2024*). The 2× SanTaq PCR Mix contains Mg$^{2+}$, dNTPs, TaqDNA polymerase, PCR buffer, loading, and PCR enhancer.

The amplification program was as follows: pre-denaturation at 95 °C for 4 min; 30 cycles of 95 °C for 45 s, the annealing Tm (depending on primers) for 45 s and extension at 72 °C for 45 s; followed by a final extension of 7 min at 72 °C.

The amplification products were separated by electrophoresis on 6% non-denaturing polyacrylamide gels and visualized by silver staining according the previous report (*Han et al., 2024*). The bands were recorded by hand according the molecular weight for future analysis.

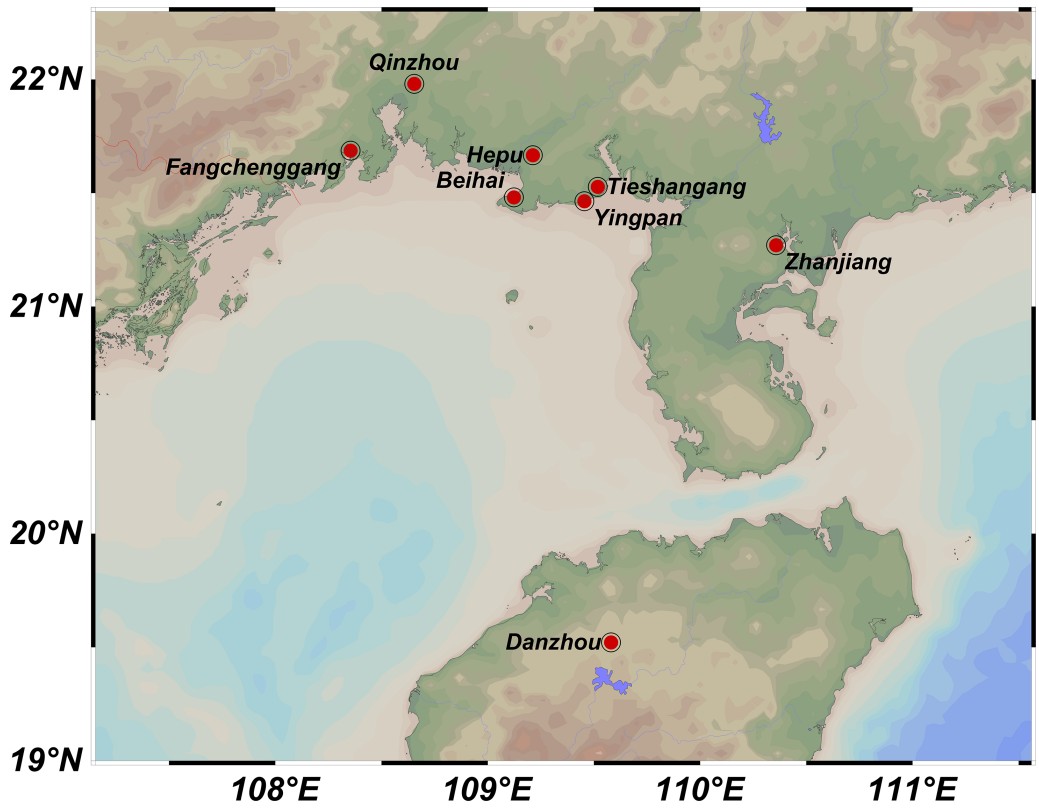

**Figure 1** **Sampling locations of *Sipunculus nudus* populations along the Beibu Gulf.** Tieshangang District, Guangxi (TSG, $n = 27$); Yingpan Town, Guangxi (YP, $n = 28$); Yinhai District, Beihai (BH, $n = 27$); Hepu County, Guangxi (HP, $n = 27$); Qinzhou, Guangxi (QZ, $n = 29$); Fangchenggang, Guangxi (FCG, $n = 27$); Zhanjiang, Guangdong (ZJ, $n = 27$); Danzhou, Hainan (HN, $n = 27$).

## Genetic diversity

GeneAlEx6.501 (*Peakall & Smouse, 2012*) was employed to calculate the parameters of genetic diversity including the number of alleles ($Na$), the effective number of alleles ($Ne$), expected heterozygosity ($He$), observed heterozygosity ($Ho$), Shannon's index ($I$), and fixed index ($F$). Additionally, Cervus V3.0 software was used to calculate the polymorphic information content ($PIC$) (*Kalinowski, Taper & Marshall, 2007*).

## Genetic distances and gene flows

GeneAIEx 6.501 was also used to assess the parameters of population genetics including gene flow ($Nm$), genetic differentiation ($Fst$), genetic distance ($Nei's$) and genetic similarity ($Nei'I$) among the populations. Molecular $AMOVA$ was performed using Arlequin V3.5.2.2 software (*Excoffier & Lischer, 2010*) to detect the distribution of genetic variation within and among populations of *S. nudus*.

DARwin6 was employed to generate a phylogenetic tree file in the format of PAU *via* neighbor-joining method with 1,000 bootstrap replications. The clustering dendrogram was drawn using the iTOL web platform.

### Genetic structure

Structure 2.3.4 software (*Sheehan et al., 2001*) was utilized to infer genetic structure using the admixture model. A burn-in period of 10,000 and Markov chain Monte Carlo (MCMC) replicates of 100,000 were set. Initially, the number of subpopulations ($K$) ranged from 2 to 10. The online software Structure Harvest was employed to determine the theoretical population number based on the $\Delta K$ algorithm (*Evanno, Regnaut & Goudet, 2005*).

## RESULTS

### Distribution and characteristics of genome-wide SSRs

A total of 277,264 SSRs (273,552 SSRs on chromosomes and 3,712 SSRs on scanffolds) was identified in the genome of *S. nudus* (1,466.684 Mb) (Fig. 2). Of them, there were 51,025 compound SSR loci. In the genome, dinucleotide motifs (213,846, 77.127%) predominated (Table 2), while pentanucleotide motifs (1,584, 0.571%) were the least. The average length of the pentanucleotides was the longest (73.045 bp), whereas that of the mononucleotides was the shortest (22.722 bp). The density was 189.041 SSRs/Mb in the genome.

There were 175,735 SSRs consisting of motifs more than 16 times, accounting for 63.382%. The SSRs with the motifs of nine times were relatively fewer (Table 3).

There was a total of 135 different motifs in the genome. Of them, the AT/AT motif dominated (170,807, 61.604%).

### Screening of polymorphic SSRs

A total of 198,902 primer pairs were successfully designed. Among them, 100 primer pairs were selected for synthesis and screened. The results showed that five primer pairs did not amplify bands, while 95 primer pairs could amplify bands including two primer pairs with single band (Fig. 3). This suggested that the successful rate of amplification was 95% and polymorphic rate was 93%.

### Genetic diversity

Thirty highly polymorphic primer pairs were employed to evaluate the genetic diversity of 219 samples of *S. nudus*. The results (Table 4) indicated that 370 alleles were detected and the *Na* ranged from five to 23 with an average of 12.333; the *Ne* varied from 2.432 to 13.743 with an average of 6.950; the *Ho* ranged from 0.758 to 1.000 with its average of 0.905; the *He* ranged from 0.589 to 0.927 with an average of 0.833; the *I* ranged from 1.050 to 2.783 with an average of 2.042; the *PIC* ranged from 0.506 to 0.923, with an average of 0.811. All these findings suggested that the genetic diversity of the population was relatively high in this study.

As for the genetic diversity of eight populations, the results (Table 5) indicated that the ZJ population had the maximum genetic diversity with the parameters of 6.233, 4.271 and 1.479 in *Na*, *Ne* and *I*. In contrast, the YP population had the minimum values of 1.933, 1.596, 0.558, respectively. The *Ho* ranged from 0.382 (YP) to 0.94 (QZ) with a mean of 0.790. The *He* ranged from 0.312 (YP) to 0.718 (ZJ) with the mean of 0.590. The values of *Ho* were more than that of *He* in all population. All the fixation indices ($F$) of the populations were negative with the mean of −0.373. According to the *He* values, the eight populations in descending order were YP, HN, QZ, HP, FCG, BH, TSG, and ZJ.

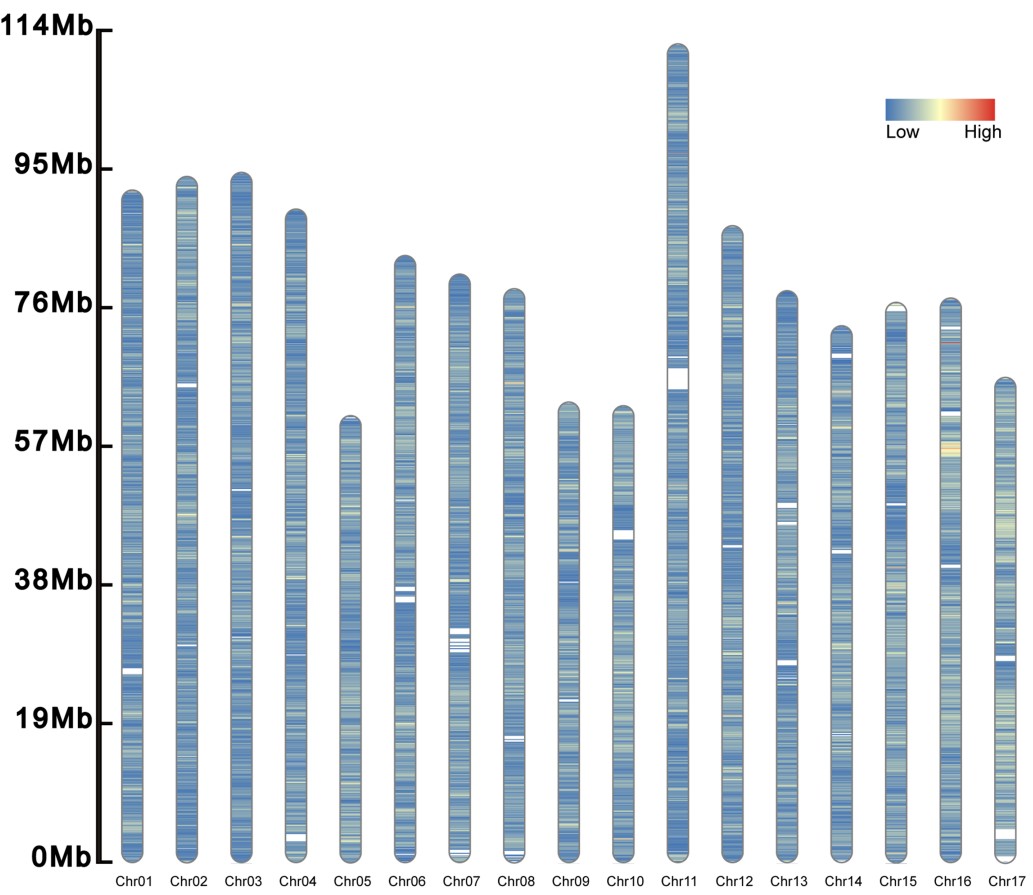

**Figure 2** **Distribution of SSRs along the chromosomes.** The SSRs on the 17 chromosomes are evenly distributed, and their densities are distributed from low to high and from blue to red.

**Table 2** **Distribution of SSRs with different repeat size in the genome of *S. nudus*.**

| Repeat unit | Number | Proportion (%) | Total length/Kb | Average length /bp |
|---|---|---|---|---|
| Mono- | 19,785 | 7.136% | 449.561 | 22.722 |
| Di- | 213,846 | 77.127% | 8,793.850 | 41.122 |
| Tri- | 21,376 | 7.710% | 693.411 | 32.439 |
| Tetra- | 18,773 | 6.771% | 495.744 | 26.407 |
| Penta- | 1,584 | 0.571% | 62.155 | 39.239 |
| Hexa- | 1,900 | 0.685% | 138.786 | 73.045 |
| Total | 277,264 | 100.00% | 10,633.507 | 38.352 |

## Analysis of molecular variance analysis

The results of Analysis of molecular variance (AMOVA) analysis showed that 3.830% of the genetic variation existed among populations, while 96.170% were observed within the population. The genetic variation mainly existed within the population rather than that between the populations. The fixation index was 0.0383.

**Table 3  Distribution of SSR repeats in the genome.**

| Repeat type | Repeat number | | | | | | | | | | | |
|---|---|---|---|---|---|---|---|---|---|---|---|---|
| | 5 | 6 | 7 | 8 | 9 | 10 | 11 | 12 | 13 | 14 | 15 | ≥16 |
| Mono- | 0 | 0 | 0 | 0 | 0 | 0 | 0 | 0 | 0 | 0 | 0 | 19,785 |
| Di- | 0 | 0 | 0 | 0 | 0 | 14,421 | 11,199 | 9,510 | 8,782 | 8,692 | 8,640 | 152,602 |
| Tri- | 0 | 0 | 4,959 | 3,047 | 2,379 | 2,077 | 1,776 | 1,480 | 1,185 | 975 | 783 | 2,715 |
| Tetra- | 8,549 | 4,449 | 2,281 | 1,063 | 779 | 510 | 330 | 176 | 272 | 62 | 43 | 259 |
| Penta- | 669 | 270 | 147 | 110 | 52 | 44 | 47 | 32 | 34 | 32 | 24 | 123 |
| Hexa- | 517 | 276 | 197 | 148 | 125 | 93 | 82 | 72 | 62 | 39 | 38 | 251 |
| Total | 9,735 | 4,995 | 7,584 | 4,368 | 3,335 | 17,145 | 13,434 | 11,270 | 10,335 | 9,800 | 9,528 | 175,735 |
| Proportion (%) | 3.511 | 1.802 | 2.735 | 1.575 | 1.203 | 6.184 | 4.845 | 4.065 | 3.727 | 3.535 | 3.436 | 63.382 |

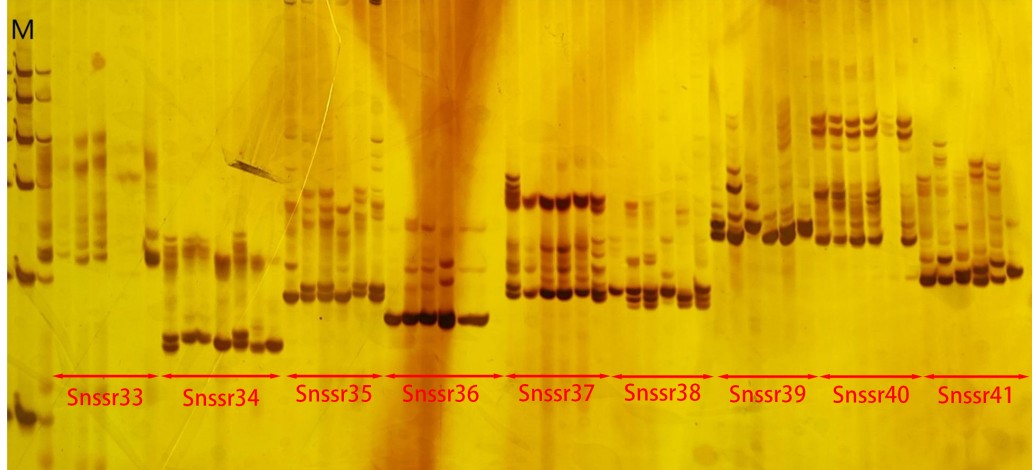

**Figure 3  The electrophoresis gel image of primer screening.** In the image, M represented Marker. The results shown were the amplification products of nine pairs of SSR primers, with their specific names labeled in the image.

## Genetic differentiation

The study revealed genetic distances between populations ranging from 0.262 to 2.998 (average 1.310), with the largest divergence between HP and HN and the smallest between HP and ZJ. Genetic similarity values spanned 0.050–0.769 (average 0.332), showing HP's lowest similarity with HN and highest with ZJ. For *S. nudus*, pairwise *Fst* values ranged from 0.049 to 0.581 (average 0.239), with the lowest *Fst* between HP and ZJ and the highest between YP and HN. *Nm* values varied from 0.180 to 4.842 (average 1.434), peaking between HP and ZJ and reaching the minimum between YP and HN.

The clustering results (Fig. 4) indicated that the samples of eight populations of *S. nudus* were divided into five groups. Group 1 contained the samples from QZ, TSG and FCG; Group 2 consisted of the individuals from HP, ZJ and BH; Group 3 included the samples from YP; Group 4 harbored the individuals from HN. Group 5 only had a single sample

Han et al. (2025), *PeerJ*, DOI 10.7717/peerj.19903

**Table 4  Genetic diversity of the population in this study revealed by 30 SSRs.**

| Loci | Left primer | Right primer | Na | Ne | Ho | He | I | PIC | F | Fst | Nm |
|---|---|---|---|---|---|---|---|---|---|---|---|
| Snssr28 | GCTTCCTTGGAAAATCAGTACA | GCCAGGCTAGACAGTAAAATAC | 7 | 5.533 | 0.959 | 0.819 | 1.76 | 0.794 | −0.17 | 0.413 | 0.355 |
| Snssr29 | ATAGAGTGCTGGGATAAGGTAC | TCAGTATAAGAAGAGGGTCACC | 5 | 2.432 | 0.895 | 0.589 | 1.05 | 0.506 | −0.52 | 0.299 | 0.585 |
| Snssr30 | CATCTGACTCACCTCTGTATCA | TGGACAGCTTTCTTGTGTTTAT | 8 | 4.508 | 0.84 | 0.778 | 1.71 | 0.746 | −0.079 | 0.330 | 0.507 |
| Snssr33 | TTTTAACTACGACAGGGCAAAA | ACGCACGGTGAACATAAAATAT | 12 | 6.568 | 0.952 | 0.848 | 2.082 | 0.83 | −0.123 | 0.332 | 0.503 |
| Snssr34 | CAATGACACCACTTCTGTTACT | TCTGTCTACTCTTCTGCTTCTT | 9 | 3.044 | 0.94 | 0.672 | 1.315 | 0.614 | −0.4 | 0.332 | 0.503 |
| Snssr35 | TGCCTGAGAACCATACCTTTAT | TCCCTTTCACCAATGGATGATA | 10 | 7.943 | 0.937 | 0.874 | 2.17 | 0.861 | −0.072 | 0.250 | 0.752 |
| Snssr36 | CGAAATGAGAAGTGGTTGTACT | CGATCGAGACATCTACAACAAA | 13 | 5.608 | 0.911 | 0.822 | 1.937 | 0.798 | −0.109 | 0.285 | 0.626 |
| Snssr37 | CCTGACCTGAAGAGTACTGTTA | GGCAGCCCTGTAATGTTAAC | 11 | 4.567 | 0.788 | 0.781 | 1.865 | 0.758 | −0.008 | 0.386 | 0.397 |
| Snssr38 | TGTGTCATGCAGGGTAGATATA | TGAGGATGAGGATGATGATGAT | 8 | 4.158 | 0.979 | 0.759 | 1.485 | 0.716 | −0.289 | 0.497 | 0.253 |
| Snssr39 | GTGTACCATTGATGTACCGAAT | GGGAATTGCTGTAGTACATAGG | 8 | 4.392 | 0.964 | 0.772 | 1.654 | 0.739 | −0.249 | 0.261 | 0.709 |
| Snssr40 | ACAGTTTGTTTCTCCTGGTCTA | CCTTAGTTTGGGAGAGGGAATA | 9 | 7.411 | 0.858 | 0.865 | 2.053 | 0.85 | 0.008 | 0.453 | 0.302 |
| Sneer41 | CCAGGGAATTGTTTGTCTGATT | GCTCCCAACTGAAAATCCATTA | 16 | 8.73 | 0.954 | 0.885 | 2.373 | 0.875 | −0.077 | 0.260 | 0.712 |
| Sneer43 | TGGTCCTCATACGTCCTATTC | AGAGAATACCTGTCTCCATGAG | 14 | 5.397 | 0.934 | 0.815 | 1.92 | 0.79 | −0.147 | 0.307 | 0.564 |
| Sneer44 | CGCTTTGTGGTAAAATGCATAA | GTTTGCCGACGGTTATATAGAT | 10 | 4.971 | 1 | 0.799 | 1.806 | 0.77 | −0.252 | 0.477 | 0.274 |
| Sneer46 | GCACAAGAACAAAAGGACATTT | ACACCTTCTTCCAATGTCTAGT | 13 | 7.242 | 0.963 | 0.862 | 2.173 | 0.848 | −0.118 | 0.415 | 0.352 |
| Sneer48 | ACTGCATAAAGTTGGCCATATC | TGCACATTGATACTAACAGACC | 15 | 7.088 | 0.898 | 0.859 | 2.183 | 0.845 | −0.046 | 0.293 | 0.602 |
| Sneer62 | ATACGCGATTGGAAAGCATAAT | ATGTGTGTGCATGTATGTATGT | 15 | 8.175 | 0.811 | 0.878 | 2.28 | 0.866 | 0.076 | 0.450 | 0.306 |
| Sneer71 | TGTTGACATCACCAGACCAAAT | ACATAGGAGCGCTGCAATAAAT | 16 | 10.261 | 0.892 | 0.903 | 2.514 | 0.895 | 0.012 | 0.152 | 1.391 |
| Sneer72 | CGCTGCCTAGTGTTACAGAATC | GTGACAATAAGTGCCGTGTTCA | 19 | 13.743 | 0.938 | 0.927 | 2.765 | 0.923 | −0.012 | 0.204 | 0.975 |
| Sneer77 | TCACTGACCGTTTGATGAATAT | TGTTCGCCATGTACGTATTT | 16 | 5.553 | 0.956 | 0.82 | 1.971 | 0.797 | −0.166 | 0.468 | 0.284 |
| Sneer79 | ACCTTTTCGAGAACCCTTAAAA | AGTGATTGTGGTGTTGGTATAG | 23 | 12.676 | 0.908 | 0.921 | 2.783 | 0.916 | 0.014 | 0.299 | 0.587 |
| Sneer83 | TGCAGGTACGTTCAATGATATA | ACTTCTTTAGCCGTGTTGTTAT | 8 | 6.165 | 0.911 | 0.838 | 1.93 | 0.818 | −0.088 | 0.309 | 0.560 |
| Sneer85 | ACAAAACAAGACACGCATATAC | CTGTCCACATTACGGTCTTTAA | 13 | 8.614 | 0.758 | 0.884 | 2.285 | 0.873 | 0.142 | 0.254 | 0.736 |
| Sneer87 | TGAAAACAAGATTGGCAGGTTA | AGGAGTCACAACAGCTGCATAC | 21 | 10.037 | 0.983 | 0.9 | 2.557 | 0.893 | −0.092 | 0.360 | 0.444 |
| Sneer90 | TGCTGAACCGGGAACAATATTA | TCGTGTGTGTGTTGGTGTATAT | 10 | 6.349 | 0.867 | 0.843 | 1.961 | 0.823 | −0.03 | 0.214 | 0.916 |
| Sneer91 | AGTTTGTGCACAGGAAGTAGAG | CCGCCGGGCCTTTATGTCTATA | 16 | 8.029 | 0.882 | 0.875 | 2.342 | 0.864 | −0.007 | 0.404 | 0.368 |
| Sneer93 | AATGCAGTGAAGCCCAGATTAT | AGACTGATGCCCTGTTCAAAGT | 12 | 8.965 | 0.828 | 0.888 | 2.295 | 0.878 | 0.068 | 0.211 | 0.937 |
| Sneer94 | TGTGCATGGTTTCTGAGTACTG | GGATTTGAGTTTGTCCTGGGGA | 12 | 8.879 | 0.967 | 0.887 | 2.29 | 0.877 | −0.09 | 0.095 | 2.390 |
| Sneer95 | ACGTGGGGTCTATGATAGTTGA | CATGGCTGCACAAGGTCTAGAA | 10 | 7.494 | 0.887 | 0.867 | 2.087 | 0.852 | −0.023 | 0.319 | 0.533 |
| Sneer97 | CACGAAAGCATCTGGTGTGAAA | CATCGCTGTCACGTTACGAATT | 11 | 3.959 | 0.786 | 0.747 | 1.661 | 0.709 | −0.052 | 0.310 | 0.556 |
| Mean | | | 12.333 | 6.950 | 0.905 | 0.833 | 2.042 | 0.811 | −0.097 | 0.321 | 0.633 |
**Table 5 The parameters of genetic diversity across provenances.**

| Populations | Na | Ne | I | Ho | He | F |
|---|---|---|---|---|---|---|
| TSG | 5.800 | 4.118 | 1.437 | 0.926 | 0.714 | −0.335 |
| FCG | 4.567 | 3.284 | 1.214 | 0.932 | 0.655 | −0.476 |
| QZ | 4.333 | 3.177 | 1.153 | 0.942 | 0.633 | −0.543 |
| YP | 1.933 | 1.596 | 0.558 | 0.382 | 0.312 | −0.210 |
| BH | 4.600 | 3.273 | 1.226 | 0.816 | 0.656 | −0.294 |
| HP | 5.133 | 3.498 | 1.262 | 0.884 | 0.650 | −0.412 |
| ZJ | 6.233 | 4.271 | 1.479 | 0.905 | 0.718 | −0.310 |
| HN | 2.300 | 1.838 | 0.674 | 0.529 | 0.379 | −0.402 |

from TSG. In Group 1, two samples from QZ appeared in samples from FCG; similarly, two individuals from HP clustered with samples from ZJ.

## Genetic structure

The results demonstrated that the $K$ with the maximum of the $\Delta K$ at the turning point was seven. Evidently, the whole population was divided into seven subpopulations in this study (Fig. 5). Subpopulation 1 included 12 individuals (TSG1∼TSG12, green); Subpopulation 2 consisted of 15 individuals (TSG13∼TSG27, orange); the subpopulation 3 (red, 56 individuals) was composed of 27 individuals (FCG1-FCG27) and 29 individuals (QZ1-QZ29); Subpopulation 4 had 22 individuals (YP1∼YP15, YP18, YP20∼YP24, and YP28, blue); Subpopulation 5 harbored 33 samples (YP16, YP17, YP19, YP25, YP26, YP27, and BH 1∼27, yellow); Subpopulation 6 (violet, 54) consisted of 27 samples from HP (HP1-27) and 27 samples from ZJ (ZJ1-27); and the samples of Subpopulation 7 come from HN (azure, HN1-27).

# DISCUSSION

## Genome-wide SSRs

Now, various types of molecular markers have been extensively utilize to the identification of marine animal genetic resources (*Kong, Li & Qiu, 2007*). To date, only few types of molecular markers including RAPD (rapid amplification polymorphic DNA) markers, DNA fragments on mitochondrial genome and SSRs (*Wang et al., 2012*) were used to evaluate the genetic diversity in *S. nudus*. However, RAPD markers had some limitations including low instability, and inability of identifying heterozygotes, while conserved mitochondrial genes offered insufficient polymorphism. High polymorphism and codominance are key features of ideal molecular markers. SSR markers, meeting these criteria, are widely regarded as the most practical. However, only few SSRs have been developed for *S. nudus* (*Guo et al., 2012*; *Wang et al., 2012*), which significantly limit molecular marker-aided breeding. To address this, large-scale SSR markers were developed in the present study.

The characterization of genome-wide SSRs revealed the presence of various repeat motifs in the genome. Among them, dinucleotide dominated (70.18%) which was consistent with the SSRs in most aquatic animal genomes such as *Protosalanx chinensis* (*Tang & Zhou,*

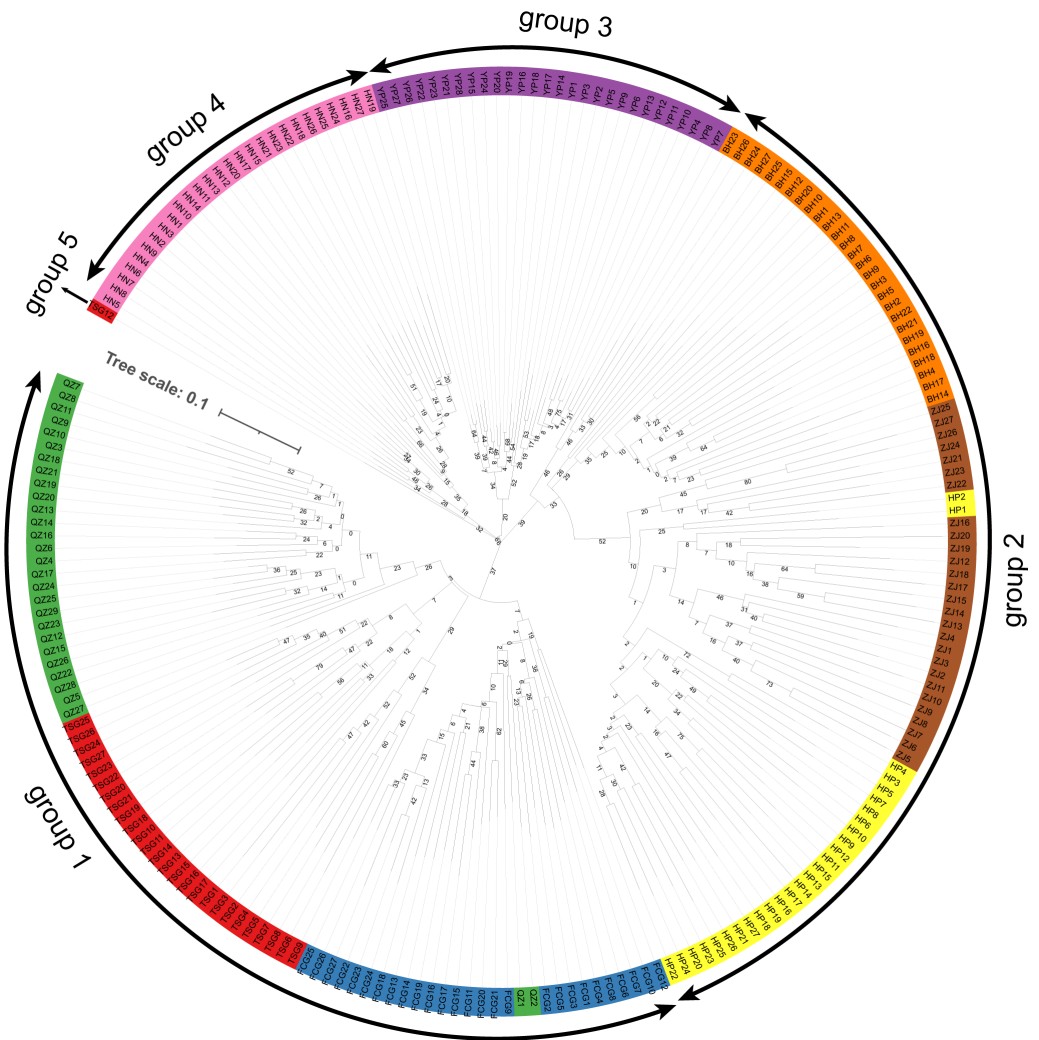

**Figure 4** **Clustering of samples of the *Sipunculus nudus*.** The green represents the Qinzhou population, the red represents the Tieshan Port population, the blue represents the Fangchenggang population, the yellow represents the Hepu population, the brown represents the Zhanjiang population, the o range represents the Beihai population, the purple represents the Yinpan population, and the pink represents the Hainan population.

*2023*), *Schizothorax biddulphi* (*Ren, 2020*), and *Puffer fish* (*Edwards et al., 1998*). Further studies revealed that the AT/AT was predominant in the genome of *S. nudus*, while AC/TG was more frequent in the genome of the *Schizothorax biddulphi*. This might be due to the differences of species and SSR criteria.

The distribution of SSRs was non-uniform, with regional variations in density and sporadic gaps detected across all chromosomes, likely attributable to taxonomic differences in SSR composition (*Srivastava et al., 2019*).

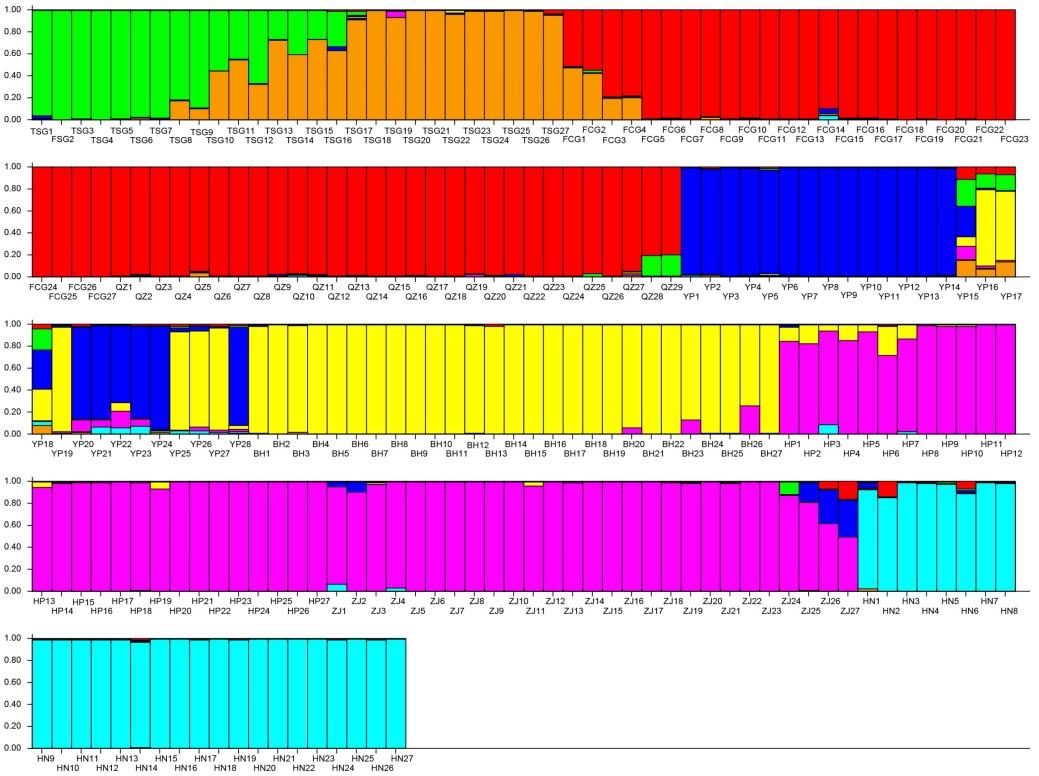

**Figure 5 Population structure of the genetic resources.** Subpopulation 1 included 12 individuals (TSG1 TSG12, green); Subpopulation 2 consisted of 15 individuals (TSG 13 TSG27, orange); Subpopulation 3 (red, 56 individuals) was composed of 27 individuals (FCG1-FCG27) and 29 individuals (QZ1-QZ29); Subpopulation 4 had 22 individuals (YP1 YP15, YP18, YP20 YP24, and YP28, blue); Subpopulation 5 harbored 33 samples (YP16, YP17, YP19, YP25, YP26, YP27, and BH 1 27, yellow); Subpopulation 6 (violet, 54) consisted of 27 samples from HP (HP1-27) and 27 samples from ZJ (ZJ1-27); and the samples of Subpopulation 7 come from HN (azure, HN1-27).

## Usability of SSRs markers

This study developed numerous codominant SSR markers and validated 100 in *S. nudus*. Of these, 95 primer pairs amplified products, with 93 exhibiting polymorphism, demonstrating their utility for assessing genetic diversity and enabling marker-assisted breeding in *S. nudus*.

## Genetic diversity

Genetic diversity, the fundamental basis of populations, forms the foundation for both environmental adaptation and genetic improvement (*Cui et al., 2012*).

The 30 polymorphic primer pairs showed *Na* and *Ne* values of 12.333 and 6.950, respectively, in the entire population—approximately four-fold higher than those reported (*Wang et al., 2012*) and three-fold greater than values documented (*Guo et al., 2012*). The average of *PIC* was 0.811, which was higher than that in other reports. All the results above suggested that the genetic diversity here was abundant (*Botstein et al., 1980*). In addition, the Shannon's index of the 30 polymorphic primer pairs in entire population were 2.042, also surpassed 0.269 and 0.394 reported previously (*Wang, Du & Li, 2006*;

*Song et al., 2011*), which may be closely related with the sample size and the types of molecular markers in another part. The results demonstrated higher genetic diversity in the population of *S. nudus* in Beibu Gulf.

In the present study, the *Ho* s were significantly higher than *He* s, which was consistent with the report (*Wang et al., 2012*). Both *Ho* and *He* were relatively similar and lower than those in this study (*Guo et al., 2012*). The observed heterozygosity exceeded the expected value, suggesting selection against homozygotes. The higher proportion of heterozygous individuals, carrying two distinct alleles with different functional roles, enhanced the population's adaptive capacity to environmental changes (*Jiang & Ma, 2014*).

## Genetic differentiation and gene flows

Genetic differentiation arises from restricted gene flow due to reproductive and geographic isolation, with key indicators like *Fst* and *Nm* quantifying divergence levels (*Ren, 2020*). Genetic differentiation levels are categorized based on *Fst* thresholds: little to none (0 <*Fst* <0.05), moderate (0.05–0.15) and high (*Fst* >0.15) between populations (*Wright & Curnow, 1978*). Therefore, moderate genetic differentiation existed among 10 pairwise populations, whose *Fst* s ranged from 0.05 to 0.15. The higher genetic divergences were among 17 pairwise populations. The genetic divergence was minimal (0.049) between ZJ and HP, while maximum (0.581) between YP and HN. The spatial segregation between the northern YP and southeastern HN populations in the Beibu Gulf restricted gene flow, leading to greater genetic differentiation. This has been confirmed (*Kawauchi & Giribet, 2014*). However, subpopulations were rarely fully isolated (occasional gene flow remains possible).

## Genetic structure

The genetic composition of a population indicates its diversity, which in turn determines a species' ability to adapt to environmental changes (*Niu et al., 2019*). Analyzing population structure through allele frequency patterns helps trace evolutionary origins, identify distinct lineages, and reveal genetic exchange between populations (*Liu et al., 2017*).

This study grouped the entire population into seven distinct genetic groups, demonstrating clear genetic structure. Moreover, the identified genetic clusters closely matched their predefined classifications.

The genetic structure is influenced by many factors including mating system, gene flow, selection and genetic drift et al. (*Wang et al., 2017*; *Hu, Chen & Francis, 2021*). Mating systems play a key role in shaping genetic structure. For *S. nudus*, sperm and eggs are released into seawater, where they are mixed evenly with current waves. This result in random fusion—a hallmark of random mating. hence, we speculated that the mating system did not drive the genetic structure in this species. This study found that both Subpopulation 5 and Subpopulation 6 consisting of various provenances showed signals of gene flow. Gene flow occurs *via* gamete dispersal (*e.g.*, ocean currents) or human-mediated transfer. Restricted gene flow over time drives population divergence, while intense market demand for *S. nudus* (food/medicine) drives overharvesting, causing population declines and potential genetic structure alterations. Our findings suggested human activities were

the primary driver of the observed genetic differentiation in this species. The observed genetic structure in *S. nudus* likely stemmed from human-driven overexploitation, which accelerated population decline and directly reduced genetic diversity *via* shifts in gene frequencies (*Cheng, Kao & Dong, 2020*). Higher observed than expected heterozygosity indicated an excess of heterozygotes. As heterozygotes carry two distinct alleles with potentially divergent functions, this suggests that populations with elevated heterozygosity may exhibit greater adaptive capacity in complex environments. Therefore, this entire population have greater adaptive capacity (*Jiang & Ma, 2014*).

## CONCLUSIONS

We firstly conducted large-scale genome-wide SSR marker identification in *S. nudus*, detecting 277,264 loci and developing 198,902 primer pairs. Of 100 selected SSR primer pairs tested, 95% successfully amplified across samples. Using 30 highly polymorphic SSRs, we highlighted the high genetic diversity, significant gene flow among provenances, as well as distinct genetic structure. This establishes a critical baseline for informed conservation and sustainable management of *S. nudus*.

## ACKNOWLEDGEMENTS

We are grateful to XiuYi Huang and Huiling Zhang (Beibu Gulf University, China) for their support in this study. We thank the anonymous reviewers for their helpful suggestions and comments on how to improve the manuscript.

### Funding

This work was supported by the Key Research and Development Project of Guangxi (GuiKe AB21220061) and independent research fund of Guangxi Key Laboratory of Marine Environment Change and Disaster in Beibu Gulf (2024ZZ04). The funders had no role in study design, data collection and analysis, decision to publish, or preparation of the manuscript.

### Grant Disclosures

The following grant information was disclosed by the authors:
The Key Research and Development Project of Guangxi: GuiKe AB21220061.
Independent research fund of Guangxi Key Laboratory of Marine Environment Change and Disaster in Beibu Gulf: 2024ZZ04.

### Competing Interests

The authors declare there are no competing interests.

### Author Contributions

- Chunli Han performed the experiments, analyzed the data, prepared figures and/or tables, authored or reviewed drafts of the article, and approved the final draft.

- Yuzhu Ni performed the experiments, prepared figures and/or tables, authored or reviewed drafts of the article, and approved the final draft.
- Guohao Yang performed the experiments, prepared figures and/or tables, and approved the final draft.
- Jialin Yang conceived and designed the experiments, performed the experiments, analyzed the data, authored or reviewed drafts of the article, and approved the final draft.
- Jie Zou conceived and designed the experiments, performed the experiments, authored or reviewed drafts of the article, and approved the final draft.
- Huijing Peng conceived and designed the experiments, performed the experiments, analyzed the data, prepared figures and/or tables, authored or reviewed drafts of the article, and approved the final draft.
- Pengliang Wang conceived and designed the experiments, performed the experiments, analyzed the data, prepared figures and/or tables, authored or reviewed drafts of the article, and approved the final draft.

## Data Availability

The genome of *Sipunculus nudus* is available at the Genome Warehouse in National Genomics Data Center: SAMC4151241.

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
