# Peer review of "Development of simple sequence repeat markers and genetic diversity of Sipunculus nudus in the Beibu Gulf of China"

_PeerJ, doi:10.7717/peerj.19903_

## Round 0.1 · original submission · Major Revisions

Dear authors, I ask you to carefully improve the manuscript and respond to the reviewers' fundamental comments. I believe that you will be able to eliminate the shortcomings and make this manuscript of higher quality.

Reviewer 1 ·

Basic reporting

Thank you for submitting this study on the genetic diversity of Sipunculus nudus. The article attempts to analyze the genetic structure and diversity of S. nudus in the Beibu Gulf by developing genome-wide SSR markers, which have potential value for resource conservation and breeding management. However, after careful evaluation, I believe the study has the following shortcomings:

1 Lack of Innovation
The study primarily employs conventional SSR marker development and genetic diversity assessment. In contrast, modern methods such as RAD-seq, WGS, RRS, and SNP analysis have become mainstream in genetic diversity and structure research. The reliance on traditional SSR markers fails to demonstrate technical advancement and cutting-edge methodologies.

2 Limited Sample Size and Geographic Coverage
Although the samples cover eight geographic locations, they are confined to the Beibu Gulf region. This limitation restricts the study’s ability to represent the genetic structure of S. nudus across a broader geographic range, reducing the generalizability of the results.

3 Lack of In-Depth Discussion of Results
Genetic diversity may be significantly influenced by environmental factors (e.g., pollution, habitat destruction), but the study does not provide any environmental background data or discuss the relationship between environmental factors and genetic structure.

4 Issues with Article Structure and Writing Quality
The introduction and discussion sections repeatedly highlight the economic and medicinal value of S. nudus and the advantages of SSR markers, making the text overly lengthy and lacking focus.
The abstract only summarizes the development of SSR markers without systematically summarizing the genetic structure of the studied populations.

5 Insufficient Background and Methodological Explanation
The introduction should include a discussion of methods used for studying genetic structure and diversity in species or explain why SSR markers were chosen for this study.
The introduction should provide a more detailed review of previous research on the genetic structure of S. nudus at the national and international levels to emphasize the importance and scientific significance of the study.

6 Unclear Interpretation of Results
For example, the statements "no significant genetic differentiation between populations (Peng et al., 2017)" and "low levels of genetic diversity among groups of S. nudus (Song et al., 2011)" need clarification. Are these findings consistent or contradictory? I understand that "no significant genetic differentiation" and "low levels of genetic diversity" are similar in meaning. Further clarification and discussion are recommended.

7 Lack of Details in Materials and Methods
Present sample information in a table and include a map of sampling locations to visually demonstrate the geographic distances between populations.
Specify the manufacturer and detailed composition or concentration of reagents like TSINGKE Master-Mix.

8 Adjustments to the Discussion Section
The first paragraph of the discussion, which describes the advantages and disadvantages of molecular markers, can be moved to the introduction to keep the discussion focused on interpreting the results.
For significant genetic differentiation observed between populations, further analysis should consider physical oceanographic factors (e.g., geographic distance, ocean currents), environmental factors, and human activities.

Experimental design

no comment

Validity of the findings

no comment

Additional comments

no comment

Reviewer 2 ·

Basic reporting

Presented manuscript and research addressed to sipunculans that are often present in many marine habitats across the Ocean. It is not surprised that this worms are interesting to many countries across the Pacific ocean.
The manuscript is written in unusual zoology style, when the object of the research presented from applied everyday side. For researchers from zoology field easy to know who is Sipunculus nudus, but for common knowledge it is strongly recommended to clarify the position of the object among the other invertebrates.
The authors place two references (Edmonds 1962 and Maxen et al 2003) and that will send the reader to the description of object, but in none of this paper is not discussed.
The other issue that raises questions is the description of the population of S. nudus with low genetic diversity. It was already shown in Kawauchi&Giribet 2013 that population across China coastline form one population, but across the World ocean five distinct lineages were recognized.
The value of this research will increase is authors will rewrite the introduction part entirely. Anyhow the present text cannot be published as present.

Experimental design

no comments

Validity of the findings

the knowledge from this research may help further not only to use the sipunculans in medicine but also to start protecting these invertebrates from human impact.

Additional comments

no

Annotated reviews are not available for download in order to protect the identity of reviewers who chose to remain anonymous.

Reviewer 3 ·

Basic reporting

I am not a native speaker myself but still text could require some polishing of the language. It is generally understandable but it sometimes not easy to read. Moreover, there are typos in the manuscript.
For example, in line 92 it is sandworm and not sandstorm.
An example of precise formulation is "by people at home and abroad" (line 89). Whose at home is it?
Additionally, while the genus name is usually abbreviated this is not the case at the beginning of the sentence. For example, in line 80 it should be Sipunculus and not S.
Moreover, the result sections 31. and 3.2 can be condensed substantially. Many of the numbers are in the tables and do not need to be repeated in the text. The text is right now more a written out table. The text can concentrate on the major aspects such as the range, or if some aspects are clearly dominate like repeats equal to or more than 16, dinucleotides or AT/AT as the dominate motif. This is sufficient. The reader can look at the tables for the other values.
In line 186, the authors state "There were 175,735 SSRs consisting of mononucleotides repeated no less than 20 times,". How can there be 175735 SSRs of repeated mononucleotides with 20 or more repetitions when there are only 19785 SSRs with mononucleotides? I assume it has to be motifs instead of mononucleotides. Additionally, in Table 2, it is more than 15 repetitions and not more than 19.
In line 192, the authors state "Twenty-three of these had more than 200 repetitions". I would not use the term repetitions here as this can be mixed up with repetitions in the SSR itself. I would rather write of loci in the genome as these are essentially the loci from before just now for the specific motif. Finally, in this paragraph, the authors conclude "The motif of AAGAT/ATCTT had the least number of repetitions (255, 0.092%)." This is only true for the 23 motifs with more than 200 loci but not for all 135 motifs. This needs to be more specific.
In the discussion, the authors refer to "artificial sandworms". I assume they mean non-native sandworms introduced from other localities.
Finally, a map with the sample localities and the sites of the populations would be very helpful to understand the distribution of the populations and which ones, for example, are close to each other or in the Tieshan Harbor area.

Experimental design

The aim of the study is "to develop a large number of genome-wide SSR markers" and therefore the authors probed the genome, designed primers for 100 loci and finally tested 30 loci.
However, two studies have already developed 16 and 18 SSR marker loci already for Sipunculus nudus making it a total of 34 and for these 34 a total of 95 loci were originally explored. Hence, these two studies together have conducted the same effort as done in this study alone. Given that there are already 34 SSR loci for S. nudus, it is not obvious what the advantage of the 30 loci in this study is comparison to the 34 that have already been developed. The introduction does not provide compelling arguments why another set of SSR loci is needed. Moreover, the discussion does not compare and address the two previous studies at all. As the three studies used different populations, the results of the three studies cannot be compared directly to each other. The different performances of the loci with respect to population genetic parameters could be just due to differences in the investigated populations and not the suitability of the loci for detecting population differences.
Considering these aspects, a proper approach to show that the 30 newly developed SSR loci are better than the already available 34 loci would be to also investigate the 34 loci for the 219 samples in this study and then compare the 30, 16 and 18 loci as well as the combination of these. This could show that the 30 new ones are better than the old ones or it could show that the old ones are already sufficient or maybe combining all 64 SSR is even better or a subset from each of the three sets is best. As off now, this study is not very different from two previous ones and only adds more SSR loci without providing any justification why they are needed.
Moreover, the authors seem to mix up the terms of primers and loci. They investigated 30 loci using 2 primers (or a primer pair). Hence, often when the authors refer to primers they mean loci. Only very rarely is it truly primers they are referring. Moreover, I assume that they mean primer pairs (forward and reverse primers) and not individual primers. For example, in the abstract, it states that they designed 226,240 primers and in line 200, they speak about 45,339 primers. It is not clear which number is correct here. Later they mention "Thirty highly polymorphic primers were employed" (line 206) or throughout the text 100 primers. From the context, it is much more likely that they mean SSR loci and not primers. The authors need to be more precise in their usage of language in this respect.
Additionally, the phylogenetic analyses can be improved or at least better displayed. A phylogram with branch length should be shown in Fig. 2 instead of only the cladogram. This will allow to assess the robustness of the results based on the length of the internal branches. Ideally, the authors would also conduct a bootstrap analysis to allow assessment of the robustness of the relationships.

Validity of the findings

See my comment above that the study lacks a comparative approach to the already published 34 SSR loci for S. nudus.

---

## Round 0.2 · Minor Revisions

Dear Dr. Wang, I kindly request you to make minor changes to this manuscript according to the reviewers' comments. I hope that this manuscript can be published in our journal as soon as possible.

Reviewer 1 ·

Basic reporting

The authors have conducted substantial work in developing genome-wide SSR markers for Sipunculus nudus (sandworm) and applying them to assess the genetic diversity and population structure in the Beibu Gulf region. The methodology is sound and the data are robust, making this study valuable for the conservation and breeding of this species. However, some minor revisions are required before acceptance. Specific comments are as follows:

1 Conclusion needs to be rewritten:The current conclusion does not adequately reflect the core findings and highlights of the study. It is recommended that the authors revise this section to clearly summarize the main results.

2 Non-italicized genus and species names:Multiple genus and species names in the manuscript (e.g., line 350) are not italicized, which does not conform to standard biological nomenclature. Please carefully proofread the entire manuscript to ensure all genus and species names (e.g., Sipunculus nudus) are properly italicized.

3 Language Editing Suggestions:
Line 1: "known as sandworm" → consider revising to "commonly known as the sandworm"
Line 61: "were so limited not to meet the demand" → should be revised to "were too limited to meet the demand"
It is suggested that the authors polish the language throughout the manuscript to improve scientific tone and avoid colloquial expressions.

4 Grammar Issues: The manuscript contains some inconsistent use of tense (e.g., “was known”--- “is known,” “demonstrated” ---“demonstrates”). A full read-through is recommended to unify tense and correct grammatical inconsistencies.

Recommendation: Minor Revision

Experimental design

The experimental design is appropriate and well-structured, effectively supporting the study’s objectives of SSR marker development and genetic diversity assessment in Sipunculus nudus.

Validity of the findings

The findings are valid and supported by comprehensive data analysis

Reviewer 2 ·

Basic reporting

The authors add a lot of improvements. The text is much better.
I would suggest only few add that are concerning to zoological field.
Line 40 The word “species” is not traditional to use it in this combination, especially for the first Sentence in Background. The use of “The marine species Sipunculus nudus, valued…” same as “earth species, Homo sapiens”. It would be better to replace it here. It maybe “marine worm”, “marine sipunculan” or at least “marine invertebrate”.

Line 77 “Sipunculus nudus, known as sandworm, is an economically important invertebrate species belonging to the genus Sipunculus in the family Sipunculidae.”
“Sandworm” is a common name in Beihai, but in common invertebrate literature this name commonly uses for specific Polychaeta worm. I would suggest specifying the use of “sandworm”.
No need to wright to point separately the genus naname as it is already present in full species name. Better to show the belonging of higher taxa.
An economically important invertebrate species, Sipunculus nudus, belonging to the family Sipunculidae, within Annelida. In Beihai it has a common name - sandworm, also peanut worm or star worm is mainly in use in literature.

Experimental design

no comments

Validity of the findings

This manuscript may be important for future research in medicine and also conservation biology.

Additional comments

no comments

Reviewer 4 ·

Basic reporting

no comment

Experimental design

no comment

Validity of the findings

no comment

Additional comments

This study scanned and identified the whole genome of Sipunculus nudus, characterized the basic features of SSR sites within the genome, developed a large number of applicable SSR markers, and further conducted a population genetics study on different populations. These results lay a solid foundation for the genetic improvement, resource management, and establishment of a germplasm resource database for Sipunculus nudus.

Currently, the authors have addressed all previously raised concerns and made substantial improvements to the language. The manuscript is now fluent overall and demonstrates logical coherence. However, the following issues remain and should be addressed. I recommend acceptance after these revisions.

Abstract
1. Line 40: A space should be added after a colon.

Introduction
2. Line 78: Please verify whether the genus name should be italicized.
3. Line 103: “mitodrial” or “mitochondrial”?
4. Line 103: Is “COI” a gene? If a gene, it needs italic.
5. Line 106: Species names should be italicized.

Materials & Methods
6. Line 131: A space should be added after a colon.
7. Please standardize the inconsistent formatting of units (Line 146: “45s”).

Results
8. Line 176: “1466.684 Mb”→“1,466.684 Mb”.
9. Have you considered including electrophoresis gel images in the Results section?

Discussion
9. Line321: Two period。
10. The Discussion section contains an excessive number of short paragraphs. Consider consolidating related content to improve conciseness and depth.

Reviewer 5 ·

Basic reporting

(1) After genetic structure analysis, it is better to carry out the principal component analysis (PCA) based on SSR markers.
(2) In the part of methods, one hundred primer pairs with the same or similar melting temperature (Tm) for the upstream and downstream primers were randomly selected and synthesized. It is not very reasonable, and a reasonable approach would be to select a certain number of primers for synthesis on each chromosome based on the proportion of SSR loci identified per unit chromosome length and the distribution characteristics of different SSR types, thereby ensuring that SSR primers can cover the entire genome completely.
(3) Some writing format errors needs to be revised in the text of word version, such as Line 78 Sipunculus is italic, Line 114 Mg2+, and so on.
(4) Lots of errors needs to be revised in the part of reference(word version), such as Line 352-253, Line 359-360, Line 362-363, Lin 366, Line 408, Line 411, Line 414, Line 423, Line 441, Line 447, Line 449-450, Line 452, and so on.

Experimental design

In the part of methods, one hundred primer pairs with the same or similar melting temperature (Tm) for the upstream and downstream primers were randomly selected and synthesized. It is not very reasonable, and a reasonable approach would be to select a certain number of primers for synthesis on each chromosome based on the proportion of SSR loci identified per unit chromosome length and the distribution characteristics of different SSR types, thereby ensuring that SSR primers can cover the entire genome completely.

Validity of the findings

Overall, the revised manuscript is of an average level.

Additional comments

No

---

## Round 0.3 · accepted · Accept

Dear Dr. Wang, I am pleased to inform you that your article has been accepted for publication. I hope that you will continue further research into sipunculids, a very interesting and important group of marine animals.

Reviewer 1 ·

Basic reporting

The manuscript presents its content clearly and is well-organized. The Introduction provides adequate context, and references are generally sufficient and relevant. Figures and tables are appropriate and informative. The authors have also provided raw data and addressed earlier formatting inconsistencies.

Experimental design

The study is a well-executed piece of original research and fits within the aims and scope of the journal. The research question is clearly defined, and the study addresses a real gap in the molecular resources available for Sipunculus nudus, an economically and ecologically important marine invertebrate.

Validity of the findings

The findings are valid and well-supported by the data.

Additional comments

This revised version of the manuscript shows clear improvement in structure, formatting, and clarity compared to the previous version. The development of a large number of SSR markers for S. nudus represents an important contribution to genetic resource development for marine invertebrates.

Reviewer 4 ·

Basic reporting

No comment

Experimental design

No comment

Validity of the findings

No comment

Additional comments

The author has answered all my questions and made the corresponding modifications, finally agreeing to accept the paper.

Reviewer 5 ·

Basic reporting

There are still some formatting errors in the references, such as Line 418 BMC plant biology, Line 441 Proto-salanx, Line 445 no doi, and so on. Please revise them carefully.

Experimental design

No comments

Validity of the findings

No comments

Additional comments

No comments